# Is Forensic Entomology Lost in Space?

**DOI:** 10.3390/insects13010011

**Published:** 2021-12-22

**Authors:** Denis R. Boudreau, Gaétan Moreau

**Affiliations:** Département de Biologie, Université de Moncton, Moncton, NB E1A 3E9, Canada; edb5864@umoncton.ca

**Keywords:** autocorrelation, blow flies, *Calliphoridae*, forensic sciences, semivariance, spatial statistics, species distribution

## Abstract

**Simple Summary:**

Although they affect most parameters that influence insect presence and development, spatial and scale effects are not normally considered in forensic entomology. Here, we documented the spatial dynamics of an important forensic taxon, the *Calliphoridae*, in the Greater Moncton area of New Brunswick, Canada. Results indicated that regional aggregation patterns of *Calliphoridae* differed among species and that the spatial relationship between species varied between aggregation and spatial anticorrelation. Overall, this study suggests that the dynamics of forensic insects in space differ in many ways among species, highlighting the importance for forensic entomology to consider spatial effects.

**Abstract:**

Spatial and scale effects have barely been considered in forensic entomology, despite their pervasive influence on most of the parameters affecting the development of insect larval stages and the progression of insect succession on cadavers. Here, we used smoothing/interpolation techniques and semivariograms to document the spatial dynamics of sarcosaprophageous *Calliphoridae*, an important forensic taxon, in the Greater Moncton area in New Brunswick, Canada. Results indicated that the spatial dynamics of *Calliphoridae* differed between species, some species showing strong patterns of regional aggregation while others did not. Multivariate spatial correlations indicated that interspecific relationships in space varied widely, ranging from local and large-scale aggregation to spatial anticorrelation between species. Overall, this study suggested that even within a restricted timescale, the spatial dynamics of *Calliphoridae* can operate at many scales, manifest in different patterns, and be attributed to multiple different causes. We stress that forensic entomology has much to benefit from the use of spatial analysis because many important forensic questions, both at the fundamental and practical levels, require a spatial solution.

## 1. Introduction

A fundamental question in forensic entomology is whether certain environmental parameters affect the ability to draw entomological inferences from insects. Thus, several parameters that are mostly abiotic in nature were extensively documented for their influence on the development of insect larval stages and on the progression of insect succession on necromass [1,2]. Spatial and scale effects, however, have barely been considered in forensic entomology [3,4,5,6,7,8], despite their pervasive effects on most of the biotic and abiotic parameters affecting insect occurrence and development. Space here refers to the place where the organisms involved in decomposition interact with each other and with the abiotic environment [9]. The area of expertise concerned with spatial and scale effects is spatial statistics [10], which provides rigorous techniques for drawing inferences about spatially distributed objects, such as cadavers in a field.

Forensic entomology has much to gain from the implementation of spatial statistics into its protocols because field data in forensic science are intrinsically composed of several layers of spatial structure. For example, at the local scale, entomological data can be influenced by spatial effects due to heterogeneity in field conditions [11,12], microclimate [13], and movement between cadavers/carcasses of organisms that can synchronize carcass dynamics (i.e., forensic species of interest, predators, parasitoids, scavengers; [14,15]). A small-scale spatial structure is to be expected if cadavers/carcasses are in close proximity to each other, as is often the case in body farms or small field sites. At larger scales, spatial effects may occur due to responses to climatic conditions [16], metapopulation and metacommunity dynamics, source-sink dynamics [17] and habitat specialization of forensic species of interest [18]. Many other situations are susceptible to producing a spatial pattern and failure to consider spatial layers can lead, for example, to underestimation of the influence of spatial dynamics on processes, to misattribution of spatial effects to other sources of variation [19], to inaccurate mechanistic inference, to erroneous projections from the data, such as when estimating the postmortem interval, etc.

The objective of this paper is to demonstrate what can be gained from the use of spatial statistics in forensic entomology. Using georeferenced data from a trapping study on Diptera of forensic interest, we map the regional distribution of species through smoothing and interpolation techniques. The latter techniques work on the premise that “Everything is related to everything else, but near things are more related than distant things” [20]. Next, we investigate autocorrelation at the population level and interspecific level using scatterplots of spatial dependence called semivariograms [21]. We then discuss what has been gained by examining the spatial dynamics of forensic species and offer some recommendations for the inclusion of spatial statistics in the field of forensic entomology.

## 2. Materials and Methods

### 2.1. Source of Data

The data used in the following analyses are from a study carried out in the Greater Moncton area in New Brunswick, Canada [22]. Briefly, nine sampling sites were established on three linear transects, each containing an urban, a periurban and a forest site. The study area was thus configured as a triangular shape (Figure 1a). The minimum distance between two sampling sites was 1.1 km. “Flies Be Gone” commercial traps modified to accommodate a bait of 60 g of minced pork liver were hung approximately 2.13 m from the ground at each sampling site to recover Diptera of forensic interest and prevent recovery of ground-dwelling arthropods. While use of bait traps as substitutes for carrion is disputed due to their inaccurate representation of carcasses/cadavers and the variability associated with the bait used [23,24], they remain efficient tools to document regional fauna of sarcosaprophageous *Calliphoridae* species of forensic interest [23,25]. Every 3–4 days from May to November 2019, traps were recovered, and a new trap was set up. Most of the collected specimens were sarcosaprophageous calliphorids (Diptera: *Calliphoridae*), also known as blow flies. Calliphorids feed on vertebrate carcasses, are often the most abundant early colonizers of this substrate in terrestrial ecosystems and are frequently used to deduce clues about a cadaver, including the postmortem interval [1,26]. Hereinafter, only *Calliphoridae* will be discussed. *Calliphoridae* specimens were identified using the current literature (i.e., [27,28,29]).

### 2.2. Statistical Analyses

Analyses were carried out using R version 4.1.1 [30]. Care was taken to ensure that data and models satisfied all assumptions of the respective analyses. The dataset of *Calliphoridae* species was separated into four subsets (early spring, late spring/early summer, late summer, and fall). These subsets corresponded to clusters where species and their abundance were more consistent based on a multivariate regression tree [31] carried out using the mvpart function and including Julian dates as a continuous factor. Here, for convenience, we modeled the four species collected in greater abundance using late summer data (i.e., Julian dates 197–256; mid-July to early-September) because the frequency of zeros is reduced in this subset and because homicides are more frequent during this period in North America [32]. These four species accounted for nearly 98% of *Calliphoridae* specimens during this period.

*Calliphoridae* abundance was log_10_ + 1 transformed for the analyses. The values obtained were mapped by inverse distance weighting (IDW) using the idw function in gstat. The formula used was:(1)Zj=∑iZidijp∑i1dijp,
where *Z_j_* and *Z_i_* represent the density estimate at any given spatial coordinate and at a sample point in space, respectively; *d_ij_* represents distance between an estimated spatial *j* point and a sample point *i*; and *p* represents the power parameter. The latter was determined for each individual species by selecting the integer with the lowest value of jackknifed root mean squared error.

Semivariograms and crossvariograms were calculated using gstat. A semivariogram [33] is half a variogram, the latter being a function describing the degree of spatial dependence of a spatial process. Semivariograms illustrate the level of statistical dependency as distance between sampling sites increases. Crossvariograms display the values of the semivariograms for the combination of two specific variables and are thus the equivalent of a multivariate spatial correlation. Positive crossvariogram values indicate autocorrelation between variables at a given distance, negative values indicate anticorrelation while null values indicate that there is no correlation structure between these variables at this distance. For each semivariogram and crossvariogram, the trend was estimated using a one-way additive model with the mgcv function.

## 3. Results

Figure 1 presents IDW estimations of the spatial distribution of four abundant *Calliphoridae*. *Calliphora livida* (Hall) exhibited greater densities on the outskirts of the study area (i.e., the corners of the triangle; Figure 1b). *Lucilia illustris* (Meigen) exhibited low densities on the westernmost longitudes and high densities on the easternmost longitudes of the study area (Figure 1c). *Lucilia sericata* (Meigen) was mostly concentrated within the center of the study area (Figure 1d). *Phormia regina* (Meigen) densities were similar to that of *L. illustris* but the former exhibited densities close to the extreme values less often (Figure 1e).

Figure 2 presents semivariograms and crossvariograms displaying the statistical dependency within and between species, respectively. The semivariograms (Figure 2a,c,f,j) indicated that *C. livida*, *L. illustris* and *P. regina* exhibited decaying statistical dependency. For *C. livida* and *P. regina*, this autocorrelation spanned the range of distances but was not very substantial (i.e., there was only a small increase in semivariance with increasing distance) (Figure 2a,j). For *L. illustris*, autocorrelation was more substantial but disappeared at half the range (Figure 2c). *Lucilia sericata* exhibited an atypical autocorrelation pattern that did not reach a plateau but instead displayed a strong autocorrelation at distances < 5 and >14 km and a weaker autocorrelation at intermediate distances (Figure 2f).

Crossvariograms were produced to display the spatial relationships between the four species of *Calliphoridae* (Figure 2b,d,e,g–i). Here, four patterns were observable: (i) no structure or association at a small distance and increasing autocorrelation with increasing distance (Figure 2b); indicates that when one species is locally abundant, the other species has high densities in distant areas; (ii) no structure or association at short and long distances but autocorrelation at intermediate distances (Figure 2e,i); indicates that when one species is locally abundant, the other species is abundant in surrounding areas; (iii) no structure or association at short and long distances but anticorrelation at intermediate distances (Figure 2d,g); indicates that when one species is locally abundant, the other species has low densities in surrounding areas; and (iv) strong positive autocorrelation at all distances (Figure 2h); indicates that when *L. illustris* is abundant, *P. regina* is generally abundant in plots of the study area; the reverse is also true.

## 4. Discussion

Space and time are the two variables at the origin of all variation because, ultimately, even abiotic variability is driven by these variables. While time has been of great concern to forensic entomologists as evidenced by the importance of research on the postmortem interval, the period of insect activity [34] and the pre-appearance interval [35], space remains an essentially unexplored dimension (but see [5,6,7,8]). To our knowledge, this study presents the first analysis of regional spatial dynamics in forensic entomology. This analysis suggested that even within a restricted timescale, spatial dynamics of forensic species can operate at many scales, manifest in different patterns, and be attributed to multiple different causes.

### 4.1. Spatial Dynamics of Calliphoridae Species

Both the IDW distributions and the autocorrelation ranges of the semivariograms differed among the four *Calliphoridae* species, suggesting that their distributions are determined by ecological factors acting at different spatial scales. The IDW distribution of *C. livida* indicated that the species was associated with the forest area on the map while the semivariance of its distribution identified by the semivariogram remained essentially constant at all distances. By contrast, *L. illustris* exhibited patterns of spatial aggregation/patchiness in both the IDW distribution (i.e., higher densities in the east of the study area) and the semivariogram (i.e., positive autocorrelation at distances < 7–8 km). Because the area of high density did not correspond to the landscape configuration (e.g., urban, periurban, or forested area; Figure 1a), it appears that mechanisms other than habitat association are structuring the spatial dynamics of this species. If we look for differentiating characteristics of these high-density sites, they all seemed to have a high degree of naturality relative to their landscape configuration. Indeed, the two urban and periurban sites most dense in *L. illustris* had nature parks and forested woodlands in their vicinity, respectively. In contrast, the other urban site did not have a nature park and the other periurban site was surrounded by grassland instead of woodland. These nearby natural environments perhaps provide resources and/or microhabitats favorable to this species. The IDW distribution of *L. sericata* showed high and contrasting densities in the central part of the study area, which is urbanized, as well as strong positive autocorrelation between close or distant sites. This autocorrelation pattern is apparent on the IDW distribution, with the closer points showing similar densities while the farther points (i.e., the three corners of the triangle) also show similar densities (Figure 1d). This is an indication that habitat specialization plays a major role in the distribution of *L. sericata*, a result that can be explained by the strong synanthropy of this species [36,37]. While temporally variable, we can speculate that there is a higher density of garbage in urban than in other habitats and suggest that studies examining the effects of garbage density on *Calliphoridae* could potentially be of ecological and forensic interest. The last of the species, *P. regina*, had a similar spatial distribution to *L. illustris* although it was more uniform and its semivariance, as identified by the semivariogram, was essentially constant at all distances. In summary, both *Lucilia* species showed an aggregated short-range distribution pattern influenced by local processes but *C. livida* and *P. regina* did not. Considering that aggregation patterns were previously detected in *Lucilia* [4], this suggests that such a mechanism may play a particularly strong role in the spatial structure of species from this genus.

Alternatively, when the semivariance of different species was combined to examine their interrelationships in space (i.e., the crossvariograms), a much more convoluted story emerged. *Calliphora livida* exhibited anticorrelation with *L. sericata* and *P. regina*, and a positive autocorrelation at long distances with *L. illustris*. In accordance, low densities of *C. livida* were documented where other species were most abundant (and vice versa) in IDW distributions. This negative spatial relationship between *C. livida* and all other species thus resulted in interspecific spatial segregation. As *C. livida* is best suited for weather conditions colder than those of late summer [22,38], the species apparently favored habitats that were not as optimal for its interspecific competitors. This demonstrates how seasonality and biotic interactions such as competition may play a role in the spatial partitioning of species [39]. The positive aggregation at the intermediate range seen on the crossvariograms of *L. sericata* with *L. illustris* and *P. regina* can be explained by the fact that the eastern zone of the abundance of the latter two species is at an approximately intermediate distance from the center of the triangle where *L. sericata* is abundant. This spatial pattern is apparently an artifact of two or more processes that independently structured each species. Habitat specialization is the most likely mechanism for *L. sericata* while other mechanisms such as resources distribution are more likely for *L. illustris* and *P. regina* (see below). The crossvariogram of *L. illustris* and *P. regina* displayed the strongest autocorrelation of all species pairs and remained so at all distances. In simple terms, this means that when one species is abundant in late summer, the other is also consistently abundant at all distances in our study area. The reverse is also true. This suggests that key factors structuring spatial/ecological distribution are shared by these two species. Although these spatial patterns are evident, their cause cannot be determined in the present analysis and further work will be required to identify the factors involved. If we venture to speculate on this phenomenon, we can hypothesize that the synchronized spatial dynamics of these two habitat generalists in our study area [22] is due to patchy resources based on two reasons: (i) their primary resource, carrion, is ephemeral and therefore dispersed in space and time [40], and (ii) they are dominant competitors in our study area [23,41]. The less pronounced spatial aggregation of *P. regina* relative to *L. illustris* may be explained by the greater ecological plasticity of the former (e.g., its ability to feed on dung [27]) and/or by the fact that it is a late colonizer of carrion in our study area compared to *L. illustris* [41]. These two mechanisms are also consistent with what the semivariograms suggested, namely that the spatial dynamics of *L. illustris* and *P. regina* operate at different scales.

### 4.2. Contribution of Spatial Statistics to Forensic Science

Most studies examining species distribution report qualitative associations of species to environmental variables that are implicitly spatial (e.g., habitat, altitude; [7,37,42]). However, the local density of a species can vary within and among habitats [6], thus rendering studies that fail to account for spatial heterogeneity potentially unreliable. Indeed, our spatial analysis indicated that processes other than simple environmental/habitat associations govern the regional distribution of all *Calliphoridae* species studied but *L. sericata*.

As was demonstrated, the two spatial analyses presented here can be useful (i) to display the spatial distribution of species, which can be informative when predicting the occurrence of *Calliphoridae* in a forensic context, and (ii) to expose patterns of species aggregation/segregation, which can uncover mechanisms not typically explored in forensic entomology. Spatial analyses lend themselves well to visualizing spatial structure, which is probably more important in the applied context of forensic entomology than the reasons for that structure since it will affect inference (e.g., cadaver relocation) regardless of underlying causes.

This study used regional data and offers little information on small-scale spatial analysis. In the experimental context of forensic entomology, small-scale effects are likely to bias results, especially if cadavers/carcasses or forensic insect traps are located close to each other. Biotic interactions (e.g., competition, predation), which can be major drivers of processes at this scale [43], can also lead to movement of species according to gradients of microhabitat suitability thus causing interspecific aggregation/segregation patterns (e.g., [44]). Such local spatial effects might even generate an artificial but consistent pattern of insect succession that would not be present in isolated carcasses. We therefore emphasize the need for small-scale spatial studies of autocorrelation in forensic entomology.

Spatial effects also extend beyond the regional scale. At larger scales, it is possible to examine whether insect species assembly is governed by the same factors in different geographic areas. Species that are highly structured by habitat variables, such as *L. sericata*, are likely to be similarly structured in other areas. Conversely, species that are not structured by strong habitat associations are likely to be more affected by metapopulation and metacommunity dynamics and source-sink dynamics. In addition, different patterns and processes operate at many scales [45], and as scale increases, the spatial non-stationarity of a given variable generally increases [46]. Taken together, these factors can contribute to greater variation, which can overrule the effects of causal relationships studied at large scales. On the other hand, when variables are chosen wisely and space is well accounted for, multi-regional studies can be more powerful in identifying causal relationships. We argue that forensic entomology would benefit from such ecological studies, which would allow for a better understanding of the relationships between species and their environment and the large-scale processes that influence the distribution of species of forensic interest.

### 4.3. Recommendations for Forensic Entomology

Based on what was reported herein, we are issuing the following recommendations:The thumb rule of using a minimum distance of 50 m between carcasses (e.g., [3,8,47,48]) to achieve some independence amongst experimental units needs to be re-evaluated using approaches documenting distance-dependent autocorrelation while accounting for time dependence associated with repeated sampling (e.g., spatio-temporal semivariograms). Although the 50 m distance limits larval movement between carcasses [15], it does not preclude synchronization of carcass dynamics through the movement of Diptera and Coleoptera adults from one carcass to another. The documentation in the current study of strong autocorrelation patterns in *L. illustris* and *L. sericata* under distances of a few kilometers indicates that the potential for this to occur is very real. The intention here is not to make forensic experiments any harder but rather to understand the consequences of the methodology used on the dynamics of decomposition and the organisms involved.Researchers should consider georeferencing experimental units at all scales (i.e., both the position of cadavers/carcasses in the field and the position of study sites for larger-scale projects). Multi-site studies cannot afford to ignore spatial and scale effects as these may be more influential than the variables under study.If researchers are not comfortable with spatial statistics, they can also choose other ways to incorporate spatial effects into their statistical models. For example, additive models lend themselves very well to the inclusion of geographic coordinates and autoregressive structures using, for example, tensor product smoothing. It is also possible to check the autocorrelation of residuals in most statistical procedures. Spatial analysis can be complex, but it is not necessary to know every facet of it to account for spatial interdependence.

## 5. Conclusions

It is often said that correlation does not imply causation. Similarly, spatial autocorrelation does not imply spatial causality because processes occurring at different scales can combine to create autocorrelation. Griffith [49] defines spatial autocorrelation as “a surrogate for unobserved geographical variables.” Thus, spatial analysis is not an end but rather a means to generate new hypotheses and better decipher the behavior of a system. In this sense, forensic entomology has much to benefit from the use of spatial statistics because many important questions, both at the fundamental and practical levels, require a spatial solution. As evidence of this, it is useful to note that other fields of forensics (e.g., [50,51,52]), the field of criminology (e.g., [53]), and other fields studying decomposition (e.g., [54,55,56]) have already adopted spatial methods.

## Figures and Tables

**Figure 1 insects-13-00011-f001:**
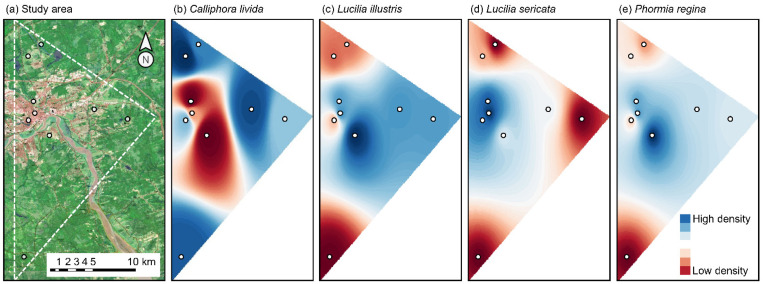
Map of study area (**a**) and estimated density (**b**–**e**) for the four most abundant *Calliphoridae* species during late summer 2019 in the Greater Moncton area, New Brunswick, Canada. Sampling sites are illustrated using white dots. Data used to generate this figure are a subset of a previously published study [22].

**Figure 2 insects-13-00011-f002:**
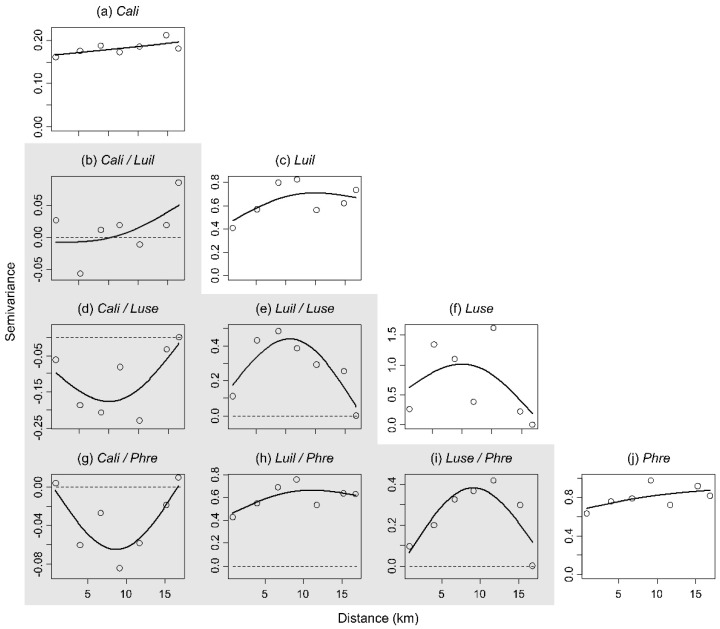
Semivariograms (**a**,**c**,**f**,**j**) and crossvariograms (gray area; **b**,**d**,**e**,**g**–**i**) for the four most abundant *Calliphoridae* species during late summer 2019 in the Greater Moncton area, New Brunswick, Canada. *Cali*, *Luil*, *Luse* and *Phre* represent *C. livida*, *L. illustris*, *L. sericata* and *P. regina*, respectively. Data used to generate this figure are a subset of a previously published study [22].

## Data Availability

The data presented in this study are available on request from the corresponding author. The data are not publicly available due to ongoing further studies and unfinished manuscripts.

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
