# Peer review of "Is Forensic Entomology Lost in Space?"

_insects, 2021, doi:10.3390/insects13010011_

Round 1

Reviewer 1 Report

This is a very original study focusing on spatial effects on Blow Flies populations, and the use of spatial statistics to analyse trends & correlations.

While this approach is both new and exciting, I have some concerns regarding this article.

First of all, I strongly suggest to consider the opportunity of changing this submission into a Technical note rather than a Study. Indeed, while the authors performed field experiments, these data are barely reported and only serve as a dataset to demonstrate the efficiency and relevance of spatial analysis. Further, the results highlighted by these analyses are trends rather than true findings. Some are not new (e.g. the synanthropy of L. sericata), others deserve complementary experiments to be confirmed. Chapter 4.3. "Recommendations for (forensic) entomology" is also clearly within the scopes of a Technical note.

Second, the authors should better discuss their field methodology (trapping), and carefully read the existing bibliography. Several studies have already been performed on Calliphorids fly distribution, and some should to be cited and analyzed here.

Finally, I am quite skeptical regarding the application of this method in the context of forensic entomology. While the authors argue it may be important to consider such spatial effects, I think they failed to clearly demonstrate how and why. On the opposite, the interest for carrion ecology and more broadly ecological study is obvious.

Beside these restrictions, I have found this article interesting and original, that's why I suggest to slightly rework it and publish it as a Technical note.

More detailed comments are reported in the annotated manuscript.

Author Response

Comment 1.1. This is a very original study focusing on spatial effects on Blow Flies populations, and the use of spatial statistics to analyse trends & correlations. While this approach is both new and exciting, I have some concerns regarding this article.

Response to Comment 1.1. We thank the reviewer for his comments on our manuscript.

Comment 1.2. First of all, I strongly suggest to consider the opportunity of changing this submission into a Technical note rather than a Study. Indeed, while the authors performed field experiments, these data are barely reported and only serve as a dataset to demonstrate the efficiency and relevance of spatial analysis. Further, the results highlighted by these analyses are trends rather than true findings. Some are not new (e.g. the synanthropy of L. sericata), others deserve complementary experiments to be confirmed. Chapter 4.3. "Recommendations for (forensic) entomology" is also clearly within the scopes of a Technical note.

Response to Comment 1.2. We agree with the reviewer that this submission may be best suited to be published as a technical note. In fact, we originally considered submitting it as such, but "Insects” only accepts research articles and reviews. Thus, no change can be made in response to this comment.

Comment 1.3. Second, the authors should better discuss their field methodology (trapping), and carefully read the existing bibliography. Several studies have already been performed on Calliphorids fly distribution, and some should to be cited and analyzed here.

Response to Comment 1.3. Our field methodology was further discussed, especially in respect to your specific comments in the annotated pdf. We understand several studies have documented calliphorid flies’ distribution and have read the reviewers suggestions. These studies usually only examine the implicit spatial distributions of species by evaluating qualitative associations with certain spatial variables (e.g., habitat type, latitude). We, on the other hand, are showing how much more there is to gain by quantitatively analysing their explicit distributions in space. Still, we appreciate the literature provided. We included a few studies examining aspects of space and scale in some citations and discussed the points mentioned above to acknowledge this literature on distribution and identify their caveats as well as their differences with our study.

Comment 1.4. Finally, I am quite skeptical regarding the application of this method in the context of forensic entomology. While the authors argue it may be important to consider such spatial effects, I think they failed to clearly demonstrate how and why. On the opposite, the interest for carrion ecology and more broadly ecological study is obvious.

Response to Comment 1.4. We agree that this study is essentially carrion ecology. Indeed, this is one of the goals of the study, to highlight that (spatial) carrion ecology mustn’t be omitted in contexts of forensic entomology since it can provide regional distribution information useful in a forensic context and prevent erroneous causal relationships, thus strengthening inferences in forensic entomology. Many changes (mostly in Section 4.2) have been made to try to demonstrate and emphasize further the interest of spatial analysis in a forensic context.

Comment 1.5. Beside these restrictions, I have found this article interesting and original, that's why I s’uggest to slightly rework it and publish it as a Technical note.

Response to Comment 1.5. Aspects related to the transformation of the manuscript into a technical note are discussed in Response to Comment 1.2.

Comment 1.6. More detailed comments are reported in the annotated manuscript.

Response to Comment 1.6. We thank the reviewer for the extensive list of comments on the PDF and for taking the time and effort to improve our manuscript. It is greatly appreciated. All of these comments have been examined and most of them have been incorporated in the text of the revised version. In addition, these comments also resulted in Figure 1 being modified.

Reviewer 2 Report

This study by Boudreau and Moreau, investigated the importance of integrating spatial and scale effects in forensic entomology. To do this, they studied blow flies (an important forensic taxon), to show how spatial statistics/analyses could be used to portray the spatial distributions of these species or their aggregation/segregation in a given area. The paper is very well written, and the authors did address some limitations in the discussion section. The topic will be of interest to readers of Insects and therefore merits publication, after some changes and revision. General comments: The authors only modeled four species collected during late summer and provided a reasonable justification for this choice. However, it is not clear to the reader how important the four species used really are with regards to the whole late summer calliphorid data. How many flies were used? Did those 4 species represent 20%, 50 or 90% etc. of the data? Providing this information in the statistical analyses or results section would really help. In my opinion, it should be mentioned in the figure legends that the data used to generate them were retrieved from a previously published study and provide the reference number. Abstract: Replace “to learn” with something else. Maybe “benefit from the use of..”? Keywords: Please do not use capital letters for the keywords except for Calliphoridae. In-text citations: Please remove space between reference numbers inside brackets. Fix throughout manuscript. Specific comments: Line 72: (Figure 1a)…not (Fig. 1a). Please fix throughout manuscript. Lines 91: Please specify the months that were considered for the late summer dataset. Line 99: Define RMSE Line 111: For each “semivariogram and crossvariogram”. Line 122: (Figure 2a,c,f,j)..not (Figs. 2a,c,f,j). Please fix throughout manuscript. Figure 2: Please define the name abbreviations (Cali, Luil, Luse, Phre) in the legend. Line 278: See earlier comment about using “to learn” in the Abstract. Line 312: Name of journal: please add a dot after each abbreviation. Line 323: Remove comma before publication year. Line 329: Remove dot after “Forensic” Line 335: In the actual manuscript, it is “favouring” not favoring

Author Response

Comment 2.1. This study by Boudreau and Moreau, investigated the importance of integrating spatial and scale effects in forensic entomology. To do this, they studied blow flies (an important forensic taxon), to show how spatial statistics/analyses could be used to portray the spatial distributions of these species or their aggregation/segregation in a given area. The paper is very well written, and the authors did address some limitations in the discussion section. The topic will be of interest to readers of Insects and therefore merits publication, after some changes and revision.

Response to Comment 2.1. We thank the reviewer for his comments on our manuscript.

Comment 2.2. General comments: The authors only modeled four species collected during late summer and provided a reasonable justification for this choice. However, it is not clear to the reader how important the four species used really are with regards to the whole late summer calliphorid data. How many flies were used? Did those 4 species represent 20%, 50 or 90% etc. of the data? Providing this information in the statistical analyses or results section would really help.

Response to Comment 2.2. Additional information about the four species, which represented 98% of the specimens collected in that period, was added to the section of the manuscript discussing statistical analyses.

Comment 2.3. In my opinion, it should be mentioned in the figure legends that the data used to generate them were retrieved from a previously published study and provide the reference number.

Response to Comment 2.3. This was added in the figure’s captions.

Comment 2.4. Abstract: Replace “to learn” with something else. Maybe “benefit from the use of..”?

Response to Comment 2.4. This was modified as suggested.

Comment 2.5. Keywords: Please do not use capital letters for the keywords except for Calliphoridae.

Response to Comment 2.5. This was modified as suggested.

Comment 2.6. In-text citations: Please remove space between reference numbers inside brackets. Fix throughout manuscript.

Response to Comment 2.6. The spaces were removed.

Comment 2.7. Specific comments: Line 72: (Figure 1a)…not (Fig. 1a). Please fix throughout manuscript. Lines 91: Please specify the months that were considered for the late summer dataset. Line 99: Define RMSE Line 111: For each “semivariogram and crossvariogram”. Line 122: (Figure 2a,c,f,j)..not (Figs. 2a,c,f,j). Please fix throughout manuscript. Figure 2: Please define the name abbreviations (Cali, Luil, Luse, Phre) in the legend. Line 278: See earlier comment about using “to learn” in the Abstract. Line 312: Name of journal: please add a dot after each abbreviation. Line 323: Remove comma before publication year. Line 329: Remove dot after “Forensic” Line 335: In the actual manuscript, it is “favouring” not favoring

Response to Comment 2.7. All suggested changes were applied to our manuscript.

Reviewer 3 Report

This paper focused on the importance of examining spatial autocorrelation in occurrence and abundance of carrion breeding blow flies and makes a compelling case for its implementation in future studies examining arthropods of forensic relevance. Overall, I think this is a lovely paper and is greatly needed in the field of modern forensic entomology. The authors are correct in their assessment that this may be the first time such a statistical analysis has been used in this field to examine blow fly aggregation. This paper is a welcome contribution to our field and it was a joy to read. I only have a few suggestions to the authors to improve clarity.

Major Comments

One thing that I think would help this paper is a discussion about the geographic areas in which blow fly species were found in abundance. Specifically, it would be helpful to describe these sites in terms of the resources they may offer blow flies. Some of the sites were urbanized, yes, but were any close to dog parks (lots of feces), garbage facilities, gardens (lots of flowers for carb-loading and refuge), or close to highly trafficked roads/highways (potentially lots of roadkill)? Though, as mentioned in the manuscript, it may not be possible to determine causality from the spatial autocorrelation, this information can still be helpful in interpreting the data.

Minor Comments

Line 71 – change “a urban…” to “an urban”

Line 115 – place author name (Hall) in parentheses

Line 128 – change “display” to “displayed”

Line 210 – as the term colonize in forensic entomology specifically means “to lay eggs on”, this statement is incorrect. Phormia regina adults often visit feces for protein and potentially water (and likely as a site to find mates), but they do not lay eggs on feces and their larvae cannot successfully develop in feces.

Line 250 – see Perez et al. 2016

Author Response

Comment 3.1. This paper focused on the importance of examining spatial autocorrelation in occurrence and abundance of carrion breeding blow flies and makes a compelling case for its implementation in future studies examining arthropods of forensic relevance. Overall, I think this is a lovely paper and is greatly needed in the field of modern forensic entomology. The authors are correct in their assessment that this may be the first time such a statistical analysis has been used in this field to examine blow fly aggregation. This paper is a welcome contribution to our field and it was a joy to read. I only have a few suggestions to the authors to improve clarity.

Response to Comment 3.1. We are very grateful to the reviewer for the analysis of our manuscript and for his comments. Honestly, we are very enthusiastic about this first excursion into the field of spatial forensic statistics and believe that the approaches discussed in the manuscript could strengthen the field in the future.

Comment 3.2. Major Comment. One thing that I think would help this paper is a discussion about the geographic areas in which blow fly species were found in abundance. Specifically, it would be helpful to describe these sites in terms of the resources they may offer blow flies. Some of the sites were urbanized, yes, but were any close to dog parks (lots of feces), garbage facilities, gardens (lots of flowers for carb-loading and refuge), or close to highly trafficked roads/highways (potentially lots of roadkill)? Though, as mentioned in the manuscript, it may not be possible to determine causality from the spatial autocorrelation, this information can still be helpful in interpreting the data.

Response to Comment 3.2. We examined this as suggested to see what information could be gained. Our most notable observations are that 1) 2 of 3 urban sites were located close to municipal parks, and 2) 2 of 3 periurban sites had more natural surroundings. We reported this in our discussion. We also speculated that garbage density must be higher in urban habitats.

Comment 3.3. Line 71 – change “a urban…” to “an urban”. Line 115 – place author name (Hall) in parentheses. Line 128 – change “display” to “displayed”.

Response to Comment 3.3. This was modified as suggested.

Comment 3.4. Line 210 – as the term colonize in forensic entomology specifically means “to lay eggs on”, this statement is incorrect. Phormia regina adults often visit feces for protein and potentially water (and likely as a site to find mates), but they do not lay eggs on feces and their larvae cannot successfully develop in feces.

Response to Comment 3.4. We thank you for this technical point. The manuscript has been modified accordingly.

Comment 3.5. Line 250 – see Perez et al. 2016

Response to Comment 3.5. We thank you for this reference. We have now accounted for it in our recommendations.

Round 2

Reviewer 1 Report

Thank you for your reply and changes. As it is not possible to publish this study as a Technical Note (and because this question was not raised by the Academic Editor), I accept this study for publication as a Research Article